# *Phylum Firmicutes* in the Faecal Microbiota Demonstrates a Direct Association with Arterial Hypertension in Individuals of the Kazakh Population Without Insulin Resistance

**DOI:** 10.3390/ijerph21121546

**Published:** 2024-11-22

**Authors:** Gulshara Abildinova, Tamara Vochshenkova, Alisher Aitkaliyev, Aizhan Abildinova, Valeriy Benberin, Anna Borovikova, Nazira Bekenova, Balzhan Kassiyeva

**Affiliations:** 1Department of Science, Medical Center Hospital of the President’s Affairs Administration of the Republic of Kazakhstan, Mangilik El 80, Astana 010000, Kazakhstan; labgen-astana@inbox.ru (G.A.); valery-benberin@mail.ru (V.B.); milanya_s@mail.ru (A.B.); nazira.bekenova@mail.ru (N.B.); kassiyevabs@gmail.com (B.K.); 2Municipal Polyclinic N° 9, Astana 010000, Kazakhstan; abildinova@bk.ru; 3CF «Institute of Innovative and Preventive Medicine», Alikhan Bokeikhan Street, Building 1, Astana 010000, Kazakhstan

**Keywords:** gut microbiota, insulin resistance, arterial hypertension, *phylum Firmicutes*

## Abstract

The gut microbiota plays a fundamental role in the host’s energy metabolism and the development of metabolic diseases such as arterial hypertension, insulin resistance, and atherosclerosis. Our study aimed to investigate the potential role of the gut microbiota in arterial hypertension among individuals of the Kazakh population without insulin resistance. 16S rRNA gene sequencing of faecal samples from 197 Kazakh subjects was performed. Preliminary binary comparisons of the faecal microbiota composition depending on the presence of arterial hypertension and insulin resistance revealed statistically significant differences in the abundance of the *phylum Firmicutes*. Logistic regression analysis showed that only the *phylum Firmicutes* influenced hypertension risk in individuals without insulin resistance after adjusting for age, sex, BMI, fasting glucose, triglycerides, and triglyceride–glucose index. The higher the abundance of the *phylum Firmicutes* in faeces, the greater the risk of arterial hypertension (OR = 1.064 [95% CI 1.005–1.125]). Correlation analysis revealed a negative association between the abundance of the *phylum Firmicutes* and the triglyceride–glucose index, primarily driven by triglyceride levels. These findings suggest the potential role of the gut microbiota, especially the *phylum Firmicutes*, in the development of hypertension in individuals without insulin resistance.

## 1. Introduction

Arterial hypertension (AH) is a well-recognised risk factor for cardiovascular diseases, which contributes to premature morbidity, disability, and mortality in many countries, including Kazakhstan.

Numerous studies on the gut microbiota (GM) underscore its significant role in human health. The GM breaks down various nutrients and produces metabolites through bacterial activity (such as short-chain fatty acids, trimethylamine and its N-oxide, lipopolysaccharides, etc.) that, upon entering the bloodstream, can broadly impact host health. Generally, the GM is considered a stable system capable of handling disturbances and maintaining symbiosis. However, dysbiosis, which arises from imbalances and dysfunctions in the GM, is associated with many disorders, including arterial hypertension, insulin resistance, and atherosclerosis [1].

Insulin resistance (IR) is a pathological condition characterised by the reduced sensitivity of muscle, adipose, and liver tissues to insulin, leading to the inability of target tissues to adequately utilise blood glucose, inhibit lipolysis and glucose production by the liver, and stimulate glycogen synthesis. Insulin-resistant cells cannot properly respond to insulin or use glucose from the blood and the resulting disturbances in lipid and glucose metabolism stimulate the further development of metabolic inflammation and metabolic syndrome (MS) [1,2]. MS is a complex of metabolic disorders reflected in dyslipidaemia, AH, obesity, type 2 diabetes, or impaired glucose tolerance, all caused by IR and compensatory hyperinsulinaemia. The GM, through microbial metabolites, plays a fundamental role in the host’s energy metabolism and metabolic diseases. In animal models of AH based on IR, metabolic inflammation determines its development [3].

Another animal model of AH suggests its development is due to dysfunctional sympathetic gut interactions. The central nervous system, receiving signals about AH risk factors (excessive salt consumption, chronic psychological stress, smoking, alcohol, hyperglycaemia, etc.), engages the thalamus, amygdala, and brainstem in complex neural pathways, leading to the increased excitability of the sympathetic nervous system. Consequently, increased gut permeability, inflammatory status, and microbial dysbiosis develop [4]. Additionally, the GM and its metabolites are significant sources of signal transmission to the central nervous system and influence the brain through neuroimmune and neuroendocrine systems [5].

Thus, the role of the GM in the onset and progression of AH is more complex due to the numerous interacting mechanisms, both within the microbial community and in the host’s regulatory functions at the levels of immunity, homeostasis, gut barrier, and biological substance transit [6]. Specifically, there are at least five aspects of the relationship between AH and the GM: different types of GM, GM metabolites, sympathetic activation, the “microbiota-gut–brain” interaction, and the impact of diet and physical exercise [7]. Different animal models of AH suggest various aspects of interaction with the GM, but only an increase in studies involving human models can significantly contribute to the goal of most scientific research: identifying potential therapeutic targets.

The fact that the study of the population genetics of the GM is still in its early stages adds dynamism to the current research process [8]. For instance, radical differences in the gut metagenome between different populations have been identified [9]. Specifically, Kazakhstan’s GM differs from both its European and East Asian counterparts, although it is like the GM of other Central Asian peoples [10]. This likely affects the efficacy of bacterial metabolites in different populations, which is significant for developing and treating AH. This study aims to investigate the possible correlation between the species composition of the faecal microbiota (FM) and AH in the Kazakh population.

## 2. Materials and Methods

### 2.1. Study Design and Patient Selection

This “case/control” study continues the scientific project “Study of the Species Composition of GM in IR in the Kazakh Population”. To implement this, participants with AH were recruited to form a group of 197 study participants—73 women and 124 men—followed by stratification based on AH and IR.

Participants were selected from the Kazakh population in their third generation, aged 38–58 years, who visited the outpatient clinic of the Medical Center Hospital of the President’s Affairs Administration of the Republic of Kazakhstan for preventive purposes in 2023. All participants had similar occupations (civil servants), lifestyles, and environmental factors, potentially reducing the influence of external factors on the study’s results. Patients with chronic somatic and autoimmune diseases, cancer, those who were pregnant, individuals of other ethnicities, and those who had received antibiotics in the past 3 months were excluded from the study.

All participants completed a questionnaire. Ethnic affiliation was indicated by the participant in the questionnaire according to the item “I identify myself, my biological parents, as well as my biological grandparents on both the maternal and paternal sides as ‘Kazakh’”.

Body weight and height were measured for each participant using standardised measurements, and body mass index (BMI) was calculated according to the standard formula BMI = weight (kg)/height (m)^2^.

For the diagnosis of AH, the 2023 European Society of Hypertension guidelines were followed [11]. AH in study participants was identified through repeated measurements at intervals of 1 to 4 weeks during office visits, with systolic blood pressure (SBP) ≥ 140 mm Hg and/or diastolic blood pressure (DBP) ≥ 90 mm Hg. The result was the average of the last two out of three readings for each visit, taken with the patient seated. Secondary AH (due to renal artery disease, parenchymal hypertension, endocrine hypertension, congenital heart defects, etc.) was excluded. The diagnosis was confirmed by 24 h ambulatory blood pressure monitoring (ABPM).

IR was calculated using the triglyceride–glucose (TyG) index method, where TyG = Ln [fasting glucose (mg/dL) × fasting triglycerides (mg/dL)]/2. A TyG index greater than 4.48 was defined as indicative of IR.

Blood samples for biochemical analysis were collected from the antecubital vein in a procedure room after a 12 h fasting period for all study subjects. Plasma was separated by centrifugation at 1000× *g* (4 °C) for 10 min and stored at −30 °C for subsequent biochemical analysis. After centrifugation, serum was used for analysis on the day of blood collection. Serum glucose levels were determined using the glucose–hexokinase method on an Abbott Architect 8000 automated biochemical analyser (Abbott Laboratories, Chicago, IL, USA). Serum triglycerides (TG) were measured spectrophotometrically using the same Abbott Architect 8000 analyser.

The species composition of FM in study participants was analysed using targeted semiconductor sequencing of the 16S rRNA gene of microorganisms with next-generation sequencing (NGS) and PIC IonReporter software. A combination of two primer pools was used to identify a broader spectrum of bacteria based on sequence data.

### 2.2. Sample Collection and DNA Analysis

Sample collection was carried out by the study participants. Each participant was provided with a stool sampling kit, cotton swabs, and sterile paper towels; samples were then frozen.

Total DNA was extracted from 250.0 mg of a homogenised wet faecal sample using the PureLink™ Microbiome DNA Purification Kit (ThermoFisher Scientific, Waltham, MA, USA) according to the manufacturer’s instructions. 16S rRNA gene sequencing was performed on a next-generation semiconductor sequencer, the Ion GeneStudio S5 Plus (ThermoFisher Scientific, USA), at the Personalized Genomic Diagnostics Laboratory of the Medical Centre of the Administration of the President of the Republic of Kazakhstan (BMTS UDP). DNA libraries (a collection of nucleotide sequences of genomic DNA from the samples) were prepared following the Ion 16S™ Metagenomics Kit protocol (ThermoFisher Scientific, USA). Library preparation involved several steps:

1. PCR Product Generation: Amplification of the hypervariable region of the 16S rRNA gene, followed by cleaning and quantifying the PCR product.

2. Library Preparation: Ligating with barcoding and cleaning the ligated adapter libraries.

3. Sequencing: Sequencing of the amplified fragments on the Ion PGM™ system, followed by bioinformatics analysis using Ion Reporter™ software, including the Ion 16S™ Metagenomics Kit module.

4. Data Analysis: Statistical analysis of results, including deviation calculations (mean ± standard deviation) using the Ion 16S™ Metagenomics Kit.

All statistical evaluations were two-sided, with *p*-values < 0.05 considered statistically significant. Statistical analysis was conducted using SPSS 24.0 statistical software (SPSS Inc., Chicago, IL, USA). Statistical results were presented as mean ± standard deviation for continuous data and n (%) for categorical data. Comparisons of quantitative measures were made using Student’s *t*-test for independent samples.

## 3. Ethics

This study adheres to the principles of the Declaration of Helsinki 1964 and Good Clinical Practice. It was approved by the Local Ethics Committee of the Medical Center Hospital of the President’s Affairs Administration of the Republic of Kazakhstan (Local Ethics Committee Protocol No. 31/2022 dated 4 April 2022) and conducted under its recommendations. All participants were informed and provided written consent to participate in the study and to use their data for research and educational purposes while ensuring the confidentiality of personal information.

## 4. Results

The comparative characteristics of the studied indicators for the 197 participants are presented in Table 1. A comparative analysis of the data was conducted based on the presence of AH, irrespective of IR. The average age of participants with AH was statistically significantly higher than that of participants without AH (*p* < 0.001). There were no significant differences between the groups regarding gender (*p* = 0.58). Both groups were predominantly male. The mean values of BMI, triglycerides (TG), glucose, and the TyG index were significantly higher in participants with AH (*p* < 0.001, *p* = 0.01, *p* < 0.001, *p* = 0.001, respectively). When examining the faecal microbiota (FM) by the average content of the *phyla Actinobacteria*, *Bacteroidetes*, *Firmicutes*, *Proteobacteria*, *Verrucomicrobia*, *Spirochaetes*, and *Lentisphaerae*, no statistically significant differences were found between participants with and without AH, irrespective of IR (Table 1).

A comparative analysis of the data based on the presence of AH in the context of IR was also conducted. The average age of participants with AH was statistically significantly higher than that of participants without AH (*p* = 0.001). The ratio of men to women was virtually the same in both groups, with no statistically significant differences in gender (*p* = 0.97). Both groups were predominantly male. The mean values of BMI and glucose were higher in participants with AH compared to those without AH (*p* = 0.01, *p* < 0.001, respectively). However, the mean values of TG and the TyG index did not differ significantly depending on the presence of AH (*p* = 0.82 and *p* = 0.06, respectively). When examining the FM by the average content of the *phyla Actinobacteria*, *Bacteroidetes*, *Firmicutes*, *Proteobacteria*, *Verrucomicrobia*, *Spirochaetes*, and *Lentisphaerae*, no statistically significant differences were found between participants with and without AH in the context of IR (Table 1).

The results of the correlation analysis showed a negative correlation between triglycerides, glucose, the triglyceride–glucose index, and the number of *phylum Firmicutes* (Table 2). According to the analysis, a decrease in triglyceride levels may be associated with an increase in the number of *Firmicutes* or a decrease in bacteria may be associated with an increase in triglyceride levels. A similar correlation was found for glucose, the triglyceride–glucose index, and the *phylum Firmicutes*. No correlation with *Firmicutes* was found for age and BMI.

A strong correlation was observed between the triglyceride–glucose index and triglycerides, while a moderate correlation was found between BMI and glucose (Table 2). The identified correlation suggests that an increase in the triglyceride–glucose index is associated with an increase in these levels.

The results of the correlation analysis in the group of patients with IR showed a weak negative correlation between glucose levels and the number of *phylum Firmicutes*. No correlation was found between *Firmicutes* and the other indicators. As for the correlation of *phylum Firmicutes* in the group of patients without IR, no correlation was observed with any of the indicators (Table 3).

Given that preliminary binary comparisons revealed statistically significant differences in the content of the *phylum Firmicutes* based on the presence of AH among participants without IR, further calculations were performed using logistic regression. This analysis included *Firmicutes* content along with several commonly accepted parameters.

The impact on the development of AH was evaluated for variables such as age, gender, BMI, glucose, and *Firmicutes*. The analysis revealed that among all the variables, only *Firmicutes* showed a significant direct association with AH. Specifically, a higher quantity of *Firmicutes* was associated with an increased risk of AH (OR = 1.064 [95% CI 1.005–1.125]) in participants without IR. These results are presented in Table 4.

## 5. Discussion

This study’s main finding was identifying a significant association between the AH of Kazakh participants without IR and the average content of *Firmicutes* in the faecal microbiota.

In a healthy gut ecosystem, the *phyla Bacteroidetes* and *Firmicutes* are the dominant bacterial groups. Their abundance, ratio, and functions evolve through competitive interactions within the microbial community, under the control of the host’s immune system, physiological homeostasis, barrier functions, and transit [6].

The *phylum Firmicutes* encompasses a broad spectrum of Gram-positive anaerobic bacteria, including several species that are considered opportunistic pathogens, such as *Staphylococcus aureus* and *Clostridia*, while the *phylum Bacteroidetes* represents a group of the most prevalent and stable Gram-negative anaerobic bacteria in the GM. They play a crucial role in the metabolism of dietary fibre and polysaccharides, serving as a primary source of short-chain fatty acids (SCFAs), particularly acetate and propionate. These SCFAs influence gut mucosal integrity and immune defence [12]. In a balanced state, *Bacteroidetes* dominance typically indicates a healthy microbiota [13].

The causes of AH in humans are multifactorial. Acting in different combinations and sequences, they trigger three main pathogenic processes: sympathetic nervous system hyperactivity, renin–angiotensin–aldosterone system activation, and endothelial dysfunction [13,14]. It is likely that by either promoting or hindering the consolidation of these pathological processes, the GM plays a biological role in hypertension. Studies involving animal models without IR suggest the crucial role of dysfunctional sympathetic interactions in the gut due to high sympathetic nervous system activity [4]. Unfortunately, we could not find information on studies of the GM in individuals without IR. However, we interpret the observed expansion of *Firmicutes* as an indication of changes in the metabolic profile of the GM associated with AH for the following reasons.

An increase in *Firmicutes* relative to *Bacteroidetes* increases the risk of neurogenic hypertension [15]. Previously, such a correlation was observed in animal models of hypertension [16] and studies involving Chinese participants, but without considering the presence of IR [17]. In humans, the expansion of *Firmicutes* in the context of hypertension was less pronounced compared to animal models but remained significant as a risk factor for cardiovascular diseases [18]. According to a meta-analysis of 19 studies, it was suggested that patients with AH may have gut microbiome dysbiosis. In hypertension models, the structure and composition of the microbiota are generally characterised by an increase in *Firmicutes* over *Bacteroidetes*, as well as a decrease in microbial richness and diversity [19]. Chen J et al. demonstrated that in mice with artificially blocked synthesis of the adaptive lymphocyte protein (LNK), *Firmicutes* species predominated in the GM. LNK is a specific regulatory protein necessary for the proper functioning of immune-competent tissue and endothelial cells. Its deficiency sharply increases the concentration of angiotensin II, which raises blood pressure [20].

Human studies, despite their limited power and lack of independent replication, support these findings [21,22]. Yan Q et al., through quantitative metagenomic analysis, characterised the faecal microbiota of 60 patients with AH and 60 healthy individuals. A significant dysbiosis of the GM was demonstrated in patients with AH. The functional characteristics of the dominant microorganisms in AH were characterised by more active membrane transport, lipopolysaccharide biosynthesis, and steroid degradation. Enzymes of gut microbes involved in the production of trimethylamine (TMA) were enriched, while enzymes producing SCFAs were depleted [22]. However, in a study by Zhu L et al., at the *phylum* level, the relative abundance of *Firmicutes* did not significantly differ between participants with AH and healthy controls [23].

The activity of anaerobic gut bacteria in converting choline to TMA promotes methanogenesis and is associated with a variety of diseases [24]. Genes responsible for converting choline to TMA have been identified in various *Firmicutes* species. These bacteria are the primary producers of TMAs, compounds that can enhance vascular inflammation and the production of reactive oxygen species [22,25]. The pro-inflammatory and vasoconstrictive functions of TMA, which contribute to the development and progression of AH, have been noted by Tokarek J. et al. [26]. Guasch-Ferré M et al., in a prospective study involving 980 individuals, found that plasma metabolites produced from choline metabolism were associated with an increased risk of cardiovascular diseases [27]. The increased presence of *Firmicutes* in our study suggests an association with AH, potentially reflecting the influence of elevated TMA production.

Disruption of the production and absorption of SCFAs reduces their effectiveness in combating AH. By modulating regulatory cells, SCFAs reduce vascular dysfunction. They perform anti-inflammatory functions, modulate lipid metabolism, and regulate gluconeogenesis. Mizoguchi et al., in their study involving 377 human samples from the Japanese population, highlighted the crucial role of the GM in activating the renin–angiotensin–aldosterone system (RAAS) during sympathetic nervous system hyperactivity. In particular, the abundance of certain *Firmicutes* species was significantly higher in participants with elevated aldosterone levels. While the causal relationship has not yet been clarified, it is suggested that the GM may play a role in regulating RAAS activity through SCFAs [28].

Increased energy extraction from food contributes to obesity and metabolic inflammation. *Firmicutes* are more efficient at extracting energy from food compared to *Bacteroidetes*, thereby promoting more effective calorie absorption and subsequent weight gain [29]. It is suggested that an increased ratio of Firmicutes to Bacteroidetes reflects habitual dietary patterns rather than obesity itself. In the adult population of Ukraine, individuals with obesity exhibited significantly higher levels of *Firmicutes* and lower levels of *Bacteroidetes* compared to adults with normal weight and lean individuals [30]. The observed increase in *Firmicutes* in our study may indirectly indicate developing secondary metabolic changes.

It is suggested that the Asian GM is the most diverse and healthiest [31]. In 2015, Kushugulova A. et al. conducted a metagenomic analysis of the GM in 84 Kazakhs. Comparing GM composition data with other populations allowed them to identify specific features of the Kazakh cohort. The most striking difference was the stable predominance of the enterotype rich in *Prevotella*, which potentially reflects regional diet and lifestyle. Participants with metabolic syndrome, in contrast to the control group, had a significantly lower *Firmicutes/Bacteroidetes* ratio and *Bifidobacteria/Subdoligranulum* ratio, with an increased relative content of *Prevotella* [10].

Our findings on the pathological significance of the *phylum Firmicutes* in AH without IR, a key factor in metabolic syndrome, are consistent with current knowledge and reflect the specific features of the Kazakh cohort. The absence of IR suggests that changes in the GM metabolic profile, resulting from sympathetic nervous system hyperfunction, may predispose the subsequent development of IR [4]. An imbalance in the sympathetic nervous system, disrupted at the level of brain structures, triggers hypertension, increases intestinal permeability, and disturbs the transit of biological substances. GM dysbiosis and metabolic inflammation contribute to the further progression of AH and the development of cardiovascular diseases [32]. Further research is needed to clarify the exact mechanisms and consequences of these findings.

This study has several limitations. Firstly, the small sample size may complicate the establishment of specific relationships. Secondly, this study used faecal material, which is easily accessible but does not fully reflect the microbiome content throughout the entire digestive system. Faecal sampling was performed under uncontrolled conditions, which may also have affected the microbiome analysis. Thirdly, due to the design of our cohort study, we were unable to exclude the influence of unmeasured factors. However, our approach aligns with the goals and interests of observational research. For the same reason, we did not assess the fact that gut function depends on the interaction of various microbial species. Moreover, the cross-sectional nature of our data does not allow us to conclude causal relationships, which was not the aim of our study. In this regard, our hypotheses regarding microbial influence on the development of AH primarily point to the need for further prospective studies. Nevertheless, despite significant limitations, we suggest that the relationship between GM and AH in Kazakhs without IR may complement existing knowledge and initiate new research in this area.

## 6. Conclusions

The average abundance of the phylum Firmicutes was significantly associated with AH. These findings highlight the potential importance of the gut microbiota, particularly *Firmicutes*, in the pathophysiology of hypertension in the absence of insulin resistance.

## Figures and Tables

**Table 1 ijerph-21-01546-t001:** Comparative characteristics of indicators between patients based on hypertension status.

Indicators	Overall Group (IR+, IR−)			IR+			IR−		
	HT+ Me (Q_1_, Q_3_)	HT− Me (Q_1_, Q_3_)	*p*	HT+ Me (Q_1_, Q_3_)	HT− Me (Q_1_, Q_3_)	*p*	HT+ Me (Q_1_, Q_3_)	HT− Me (Q_1_, Q_3_)	*p*
Age	52 (49–57.8)	48 (41.3–55)	<0.001	53.5 (49–58)	48 (42–55)	0.001	50.5 (48.5–53.8)	48.0 (40.8–54.0)	0.07
Women	27	46	0.58	22	32	0.97	5	14	0.67
Men	41	83		30	43		11	40	
BMI	28.4 (25.1–31.9)	25.2 (22.3–28.8)	<0.001	29.2 (26.6–32.3)	25.7 (23.5–29.4)	0.001	25.9 (22.9–29.3)	24.1 (20.9–26.9)	0.09
TG	109.3 (86.9–162.4)	92.9 (64.2–139.8)	0.01	140.3 (103.5–171.1)	135.4 (104.4–176.1)	0.82	67.7 (57.3–75.9)	61.1 (51.3–71.0)	0.13
Glucose	100.8 (94.9–114.8)	94.9 (89.6–100.5)	<0.001	104.0 (97.1–118.4)	97.2 (91.8–102.6)	<0.001	92.3 (86.4–99.1)	99.6 (88.0–96.4)	0.87
TyG Index	4.7 (4.5–4.9)	4.5 (4.4–4.8)	0.001	4.84 (4.64–4.91)	4.76 (4.60–4.88)	0.06	4.41 (4.29–4.45)	4.36 (4.23–4.42)	0.11
*phylum* Actinobacteria	1.37 (0.49–2.31)	1.16 (0.43–2.54)	0.98	1.37 (0.44–2.69)	1.17 (0.34–3.20)	0.87	1.40 (0.74–1.95)	1.04 (0.44–2.50)	0.77
*phylum* Bacteroidetes	41.2 (31.2–47.5)	39.3 (31.5–50.4)	0.85	43.7 (33.1–48.8)	40.8 (32.8–54.1)	0.99	34.9 (27.3–41.3)	36.2 (29.1–47.6)	0.19
*phylum* Firmicutes	37.9 (29.6–44.3)	35.4 (28.4–43.5)	0.57	36.4 (27.8–39.8)	32.5 (26.4–42.6)	0.61	46.1 (39.2–50.3)	40.1 (31.9–44.7)	0.03
*phylum* Proteobacteria	19.5 (13.6–25.7)	18.6 (12.2–25.8)	0.58	20.8 (13.6–28.8)	20.9 (12.2–26.7)	0.82	18.6 (12.2–24.9)	17.5 (11.5–24.6)	0.60
*phylum* Vericomicrobia	0	0	0.35	0	0	0.65	0 (0–0.01)	0 (0–0.02)	0.88
*phylum* Spirochaetes	99.9 (99.7–100.0)	99.9 (99.7–100.0)	0.39	0	0	0.12	0	0	0.59
*phylum* Lentisphaerae	0	0	0.20	0	0	0.13	0	0	0.90

Notes and abbreviations: Quantitative data are presented as Me (median), Q1 (25th percentile), and Q3 (75th percentile). Comparisons between groups were performed using the non-parametric Mann–Whitney test. IR+—insulin-resistant. IR−—not insulin-resistant. HT+—with hypertension. HT−—no hypertension.

**Table 2 ijerph-21-01546-t002:** Correlation between indicators in the overall group (IR+, IR−).

Indicators	*phylum Firmicutes*	*p*	TyG Index	*p*
Age	−0.60	0.40	0.05	0.47
BMI	−0.08	0.24	0.43	<0.001
TG	−0.23	0.001	0.91	<0.001
Glucose	−0.26	<0.001	0.41	<0.001
TyG Index	−0.28	<0.001	-	-

Note: IR+—insulin-resistant, IR−—not insulin-resistant.

**Table 3 ijerph-21-01546-t003:** Correlation between indicators in patients with IR+.

Indicators	IR+		IR−	
	*phylum Firmicutes*	*p*	*phylum Firmicutes*	*p*
Age	−0.001	0.99	0.13	0.47
BMI	0.01	0.88	−0.05	0.70
TG	−0.08	0.38	−0.03	0.79
Glucose	−0.26	0.004	−0.002	0.99
TyG Index	−0.14	0.12	−0.04	0.74

Note: IR+—insulin-resistant, IR−—not insulin-resistant.

**Table 4 ijerph-21-01546-t004:** Impact of indicators on the risk of developing hypertension in patients without insulin resistance.

		95% CI for Exp (B)		
	B (SE)	Lower	Exp B	Upper
Constant	−9.568 (4.796)		0.000	
Age	0.092 (0.055)	0.986	1.097	1.220
Gender	0.390 (0.701)	0.374	1.478	5.834
BMI	0.150 (0.078)	0.997	1.162	1.353
Glucose	−0.029 (0.035)	0.907	0.971	1.041
*Firmicutes*	0.062 (0.029)	1.005	1.064	1.125

Note: B—*Filum Bacteroidetes*.

## Data Availability

The data presented in this study are available on request from the corresponding author due to the protection of primary data.

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
