# Peer review of "Phylum Firmicutes in the Faecal Microbiota Demonstrates a Direct Association with Arterial Hypertension in Individuals of the Kazakh Population Without Insulin Resistance"

_ijerph, 2024, doi:10.3390/ijerph21121546_

Round 1

Reviewer 1 Report

Comments and Suggestions for Authors

1. The abstract should be reorganised to improve clarity.

2. The analysis methods used in this article are singular. It is recommended to use multiple methods for data analysis to verify the results comprehensively.

3. A more detailed comparison with existing studies, particularly those involving human subjects, and an analysis of whether the unique background of the Kazakh population contributes to these findings would make this article more convincing.

Author Response

Dear Reviewer 1,

Thank you very much for your comments and recommendations! Thanks to your recommendations, our manuscript has drastically improved. Here are our point-by-point responses to your comments.

Moreover, major grammatical corrections and syntax adjustments were made to address the quality of the English language.

The given lines in the revised manuscript are indicated in a “simple mark-up” form of tracked changes.

Comment Number

Reviewers Comment

  1.  

The abstract should be reorganised to improve clarity.

Author’s Response

The abstract has been reorganized to be more succinct and the content has been amended to be clearer.

  1.  

Reviewers Comment

The analysis methods used in this article are singular. It is recommended to use multiple methods for data analysis to verify the results comprehensively.

Author’s Response

The correlation analyses between indicators in the overall group (IR+, IR-) and between indicators in patients with IR+ have been performed to further support our findings as well as to present more comprehensive view of your findings. Moreover, the discussion was completely rewritten in accordance with our findings.

  1.  

Reviewers Comment

A more detailed comparison with existing studies, particularly those involving human subjects, and an analysis of whether the unique background of the Kazakh population contributes to these findings would make this article more convincing.

Author’s Response

The discussion part was completely rewritten with more comparisons with human subjects, Asian population, and Central Asian population, ­– lines 63-78.

Reviewer 2 Report

Comments and Suggestions for Authors

Review report of the Manuscript ID ijerph-3218174

The article submitted for our consideration is of considerable interest. However, we note that the authors should make certain corrections in order to improve the scientific quality of the manuscript article.

Introduction section

At this level, the problem is very well posed. The introduction concludes with the research question, which is clearly underlined. Which is quite interesting.

Methodology section

In terms of methodology, we feel that the authors have taken all aspects of this study into account. However, the authors should have specified that the 197 participants included both men and women.

Results section

In the results section, we believe that the presentation of results needs to be improved. For such a scientific article manuscript, the results need to be better presented.

Discussion section

The discussion is not relevant enough. It must be improved. The results must be really discussed, showing the links between the different parts (Arterial Hypertension, Gut microbiota and Insulin Resistance). The conclusion is not very clear. The authors should point out the study's limitations.

Author Response

Dear Reviewer 2,

Thank you very much for your comments and recommendations! Thanks to your recommendations, our manuscript has drastically improved. Here are our point-by-point responses to your comments. Moreover, major grammatical corrections and syntax adjustments were made to address the quality of the English language.

The given lines in the revised manuscript are indicated in a “simple mark-up” form of tracked changes.

Comment Number

Reviewers Comment

  1.  

The article submitted for our consideration is of considerable interest. However, we note that the authors should make certain corrections in order to improve the scientific quality of the manuscript article.

Introduction section

At this level, the problem is very well posed. The introduction concludes with the research question, which is clearly underlined. Which is quite interesting.

Methodology section

In terms of methodology, we feel that the authors have taken all aspects of this study into account. However, the authors should have specified that the 197 participants included both men and women.

Author’s Response

Methodology section

The study group was specified, – the lines 50-51 now reads “…participants with AH were recruited to form a group of 197 study participants – 73 women and 124 men…”.

  1.  

Reviewers Comment

Results section

In the results section, we believe that the presentation of results needs to be improved. For such a scientific article manuscript, the results need to be better presented.

Author’s Response

The correlation analyses between indicators in the overall group (IR+, IR-) and between indicators in patients with IR+ have been performed to further support our findings as well as to present more comprehensive view of your findings. Moreover, the discussion was completely rewritten in accordance with our findings.

  1.  

Reviewers Comment

Discussion section

The discussion is not relevant enough. It must be improved. The results must be really discussed, showing the links between the different parts (Arterial Hypertension, Gut microbiota and Insulin Resistance). The conclusion is not very clear. The authors should point out the study's limitations.

Author’s Response

The discussion section and the conclusion have been completely rewritten to address your concerns.

The limitations were also pointed out, moreover, we provided contradicting results of other authors to offer more balanced comparison of our results.

Reviewer 3 Report

Comments and Suggestions for Authors

In the methodology, was the statistical analysis done only with DNA analysis?

This accommodates the results better since the information is lost in the tables, and the significant difference is not appreciated.

 Line 242 refers to this: Contradictory Evidence in the Literature: A meta-analysis of 19 studies revealed conflicting results regarding the correlation between AH and gut microorganisms. It was suggested that patients with AH might experience a dysbiosis of the GM, although high-quality studies with larger sample sizes are needed [21]. Is the only one reported on this controversy?

Author Response

Dear Reviewer 3,

Thank you very much for your comments and recommendations! Thanks to your recommendations, our manuscript has drastically improved. Here are our point-by-point responses to your comments. Moreover, major grammatical corrections and syntax adjustments were made to address the quality of the English language.

The given lines in the revised manuscript are indicated in a “simple mark-up” form of tracked changes.

Comment Number

Reviewers Comment

  1.  

In the methodology, was the statistical analysis done only with DNA analysis?

This accommodates the results better since the information is lost in the tables, and the significant difference is not appreciated.

Author’s Response

Yes, the statistical analysis was done based on the DNA analysis.

  1.  

Reviewers Comment

Line 242 refers to this: Contradictory Evidence in the Literature: A meta-analysis of 19 studies revealed conflicting results regarding the correlation between AH and gut microorganisms. It was suggested that patients with AH might experience a dysbiosis of the GM, although high-quality studies with larger sample sizes are needed [21]. Is the only one reported on this controversy?

Author’s Response

The discussion section and the conclusion have been completely rewritten to address your concerns.

Round 2

Reviewer 1 Report

Comments and Suggestions for Authors

I would like to thank the authors for their careful revisions and confirm that the manuscript is now suitable for acceptance.